# A Review of Polyhydroxyalkanoates: Characterization, Production, and Application from Waste

**DOI:** 10.3390/microorganisms12102028

**Published:** 2024-10-08

**Authors:** Luis Getino, José Luis Martín, Alejandro Chamizo-Ampudia

**Affiliations:** 1Área de Genética, Departamento de Biología Molecular, Universidad de León, 24007 León, Spain; luis.getino@unileon.es; 2Área de Bioquímica y Biología Molecular, Departamento de Biología Molecular, Universidad de León, 24007 León, Spain; jmartf12@estudiantes.unileon.es; 3Institute of Molecular Biology, Genomics and Proteomics (INBIOMIC), Universidad de León, Campus de Vegazana, 24071 León, Spain

**Keywords:** polyhydroxyalkanoates (PHAs), sustainable feedstocks, bioplastics, agroindustrial waste, microorganisms, extraction processes, cost-effectiveness, environmental impact

## Abstract

The search for alternatives to petrochemical plastics has intensified, with increasing attention being directed toward bio-based polymers (bioplastics), which are considered healthier and more environmentally friendly options. In this review, a comprehensive overview of polyhydroxyalkanoates (PHAs) is provided, including their characterization, applications, and the mechanisms underlying their biosynthesis. PHAs are natural polyesters produced by a wide range of prokaryotic and some eukaryotic organisms, positioning them as a significant and widely studied type of bioplastic. Various strategies for the production of PHAs from agroindustrial waste, such as cacao shells, cheese whey, wine, wood, and beet molasses, are reviewed, emphasizing their potential as sustainable feedstocks. Industrial production processes for PHAs, including the complexities associated with extraction and purification, are also examined. Although the use of waste materials offers promise in reducing costs and environmental impact, challenges remain in optimizing these processes to enhance efficiency and cost-effectiveness. The need for continued research and development to improve the sustainability and economic viability of PHA production is emphasized, positioning PHAs as a viable and eco-friendly alternative to conventional petroleum-based plastics.

## 1. Introduction

Since the 1960s, petrochemical plastics have gradually displaced traditionally used materials such as glass, wood, and metals across various sectors, becoming an essential part of everyday life. The low thermal and electrical conductivity, lightweight, rigidity, long lifespan, corrosion resistance, and low cost of fossil fuel-derived plastics have resulted in their large-scale use. In 2015, global plastic production exceeded 320 million tons per year [1], reaching a peak of 400.3 million tons in 2022 [2]. Only 9.6% of these petrochemical polymers originate from bio-based, recycled, or carbon capture sources. China is identified as the leading producer, accounting for 32% of global production, followed by North America and the European Union with 17% and 14%, respectively; these regions are also the primary producers of bio-based plastics, contributing 33%, 13%, and 27% of global production, respectively [3]. This trend is projected to continue increasing, with an estimated production of 460 million tons by 2030 [4].

Plastics are synthetic polymeric molecules whose properties render them suitable for a wide range of applications. The most commonly used plastics are petrochemical based, including polyethylene (PE), polypropylene (PP), polystyrene (PS), polyvinyl chloride (PVC), polyurethane (PUR), polyethylene terephthalate (PET), and polybutylene terephthalate (PBT) [5]. The versatility, malleability, and low water permeability of these materials have made them the preferred choice for packaging—particularly for food and other perishable products—as well as for use in construction materials, the medical and pharmaceutical fields, and the automotive and aerospace sectors. In these applications, plastics have contributed to technological and safety improvements, owing to their insulating properties [6]. However, the majority of these plastics eventually end up discarded in oceans or terrestrial environments [7], where their stability and resistance to degradation result in their persistence for hundreds or even thousands of years, leading to significant pollution (Table 1) [8].

Once deposited in nature, plastics begin to degrade slowly, generating microplastics and nanoplastics that are harmful to ecosystems and negatively impact animal life and human health by accumulating in tissues after ingestion, potentially leading to toxic or carcinogenic effects [1,15]. This degradation is driven by abiotic factors such as ultraviolet (UV) radiation and erosion, as well as microorganisms [16], including bacteria (e.g., *Arthrobacter*, *Pseudomonas*, *Bacillus*, *Streptomyces*, *Rhodococcus*, *Micrococcus*, *Corynebacterium*, and *Nocardia*), and fungi (e.g., *Fusarium* spp., *Aspergillus* spp., and *Penicillium* spp.), which produce various enzymes to break down polymer chains into simpler molecules [15]. However, the widespread use of plastics did not begin until the 1960s, resulting in insufficient time for nature to develop highly active enzymes capable of breaking them down effectively. Moreover, the diverse types of plastics accumulating in the environment often contain various blends with solubilizers and other chemical agents intended to modify their mechanical and physical properties. This complexity makes their degradation in natural environments particularly challenging [14].

In recent years, the search for and production of alternatives to petrochemical plastics have intensified, with a focus on bio-based polymers (bioplastics) that are less harmful to human health and the environment [17]. Bioplastics, produced from natural raw materials and of biological or partially biological origin [18], such as natural rubber or latex, have been utilized for centuries, with the first artificial biopolymer created from cellulose in the 19th century [19]. Bioplastics can be categorized into biodegradable and nonbiodegradable types [20]: biodegradable bioplastics can be broken down into simpler molecules in the environment [21], whereas nonbiodegradable bioplastics can only be recycled under industrial composting conditions [22]. Biopolymers are classified into three groups based on their origin: those derived from biomass, including polysaccharides, proteins (e.g., collagen or gelatin), and lipids [19]; those produced from intermediates derived from renewable raw materials, such as polylactic acid (PLA) derivatives; and those extracted from microorganisms, with polyhydroxyalkanoates (PHAs) being the most significant and widely produced [8,19]. Among all alternatives to petrochemical plastics, PHAs stand out as they form a family of natural polyesters produced in small quantities by a diverse range of prokaryotic and some eukaryotic organisms. These PHAs act as storage components in the form of cell granules, serving as reserve carbon sources [8]. Moreover, PHAs are biodegradable and degrade naturally to carbon dioxide and water due to the action of a variety of microorganisms [23]. For this reason, they are considered an ideal replacement for fossil-based plastics, as they offer the benefits of traditional materials while meeting all sustainability requirements throughout their life cycle [21]. However, due to their high production costs, it is crucial to advance research and development. This review will cover the characterization and applications of these polymers, their biosynthesis mechanisms, industrial production, and their extraction and purification.

The importance of continued innovation in the field of bioplastics, particularly PHAs, is highlighted in this review. It is emphasized that further research and development are required to improve the sustainability and economic viability of PHA production, thus positioning PHAs as a viable and eco-friendly alternative to conventional petroleum-based plastics. By addressing key challenges such as high production costs and energy-intensive recovery processes, this review contributes to advancing knowledge that will help reduce environmental impacts and foster the transition toward a circular bioeconomy.

## 2. Characteristics and Applications of PHAs

PHAs constitute a diverse group of biomaterials with versatile characteristics that are determined by the type and distribution of monomers within their polymeric structure. Although more than 150 hydroxyalkanoate monomers capable of producing PHAs are known, only a limited number have reached industrial maturity [6]. These PHAs are differentiated based on the number of carbon atoms in the main chain or the radical R, as indicated by the formula shown in Figure 1 [24].

These biopolyesters can be classified based on the carbon chain length (R) of their monomers: short-chain PHAs (4–5 carbons), medium-chain PHAs (6–14 carbons), and long-chain PHAs (more than 14 carbons) [25,26] (Figure 1). Some authors have proposed that mcl-PHAs (medium-chain length PHAs) include those with a carbon number between 6 and 16, whereas lcl-PHAs (long-chain length PHAs) are those with more than 16 carbons [27]. The size of their side chains imparts different properties and applications to PHAs (Table 2) [27]. Additionally, PHAs can be classified by composition into homopolyesters, which contain a single type of monomer, or heteropolyesters, which include multiple types of monomers. Heteropolyesters are further subdivided into copolyesters, terpolyesters, and quaterpolyesters based on whether they consist of two, three, or four different monomers, respectively [28]. The incorporation of various monomers to form new heteropolymers results in bioplastics with intermediate characteristics compared to the homopolymers of their constituent monomers [6].

PHAs can also be classified based on the carbon source used in their production. Accordingly, they are categorized into those synthesized from traditional carbon sources, such as sugars, vegetable oils, and animal fatty acids, which produce the hydroxy acids (monomers) forming the polymeric chains of PHAs [37]. Additionally, the use of modified carbon sources enables the creation of new polymers with unique functional groups on their side chains, thereby altering their morphology and physical properties [38,39,40]. The exploration of PHAs chemical modifications is ongoing to enhance their properties or adapt them for new applications. Chemical modifications, such as carboxylation, hydroxylation, and graft copolymerization, can modify their structure to improve characteristics such as hydrophobicity and biocompatibility, rendering them particularly useful for biomedical applications [41,42,43,44].

PHAs are recognized for their significant potential due to their unique characteristics, such as biodegradability, biocompatibility, and nontoxicity. These properties enable their application across various fields, including medicine, agriculture, and biofuel production [45,46,47,48].

In the biomedical field, PHAs are considered materials of interest due to their biodegradability and biocompatibility, allowing them to degrade naturally within the body without causing toxicity [49,50]. As a result, they are ideal candidates for applications such as controlled drug release. In this context, PHAs are capable of encapsulating drugs and releasing active compounds gradually and in a controlled manner, thereby enhancing treatment efficacy and minimizing side effects [51,52]. In tissue engineering, PHAs have been used to fabricate biocompatible scaffolds that provide structural support for cell growth and the regeneration of damaged tissues. These advancements hold great promise, particularly for the repair of bone, cardiac, or vascular tissues [53,54].

Short- and medium-chain PHAs can be esterified with methanol to produce hydroxyalkanoate methyl esters, which are compounds utilized in the production of biofuels. These esters are comparable to biodiesel, positioning PHAs as a sustainable and renewable alternative to fossil fuels. This process provides a pathway for the production of clean and renewable energy, contributing to the reduction of greenhouse gas emissions [29]. In the food-packaging sector, the FDA has approved several PHA-based polymers due to their ability to withstand high temperatures, protect against UV light, and act as effective barriers against aromatic compounds, preserving the aroma and flavor quality of packaged foods. These properties make PHAs an excellent option for replacing conventional nonbiodegradable plastics, thus enhancing sustainability in the food-packaging industry [55,56].

In sustainable agriculture, PHAs are seen as having enormous potential. Biodegradable plastic films made from PHAs can be applied to protect crops from adverse environmental factors, such as extreme weather conditions or pests, without the need for removal, as they degrade naturally. Furthermore, PHAs can be used to encapsulate fertilizers or seeds, releasing nutrients in a controlled manner and optimizing agricultural resource use [57]. These films can also function as biodegradable mulch, aiding in soil moisture conservation and reducing water consumption, critical factors in water-stressed regions. The degradation rate of these materials can be tailored to the climatic conditions and specific needs of individual crops, thus improving the efficiency of agricultural practices [58].

PHAs are versatile polymers with promising applications across various sectors, including medicine, agriculture, and beyond. Their ability to safely biodegrade, combined with their advantageous physical and chemical properties, positions them as sustainable alternatives to conventional plastics and fossil fuels. This helps to reduce dependency on these resources while improving the efficiency of industrial and agricultural processes.

## 3. Molecular Mechanisms

As previously explained, different precursors can lead to the accumulation of PHAs. Although microorganisms can assimilate these precursors from their environment for accumulation, they are also capable of synthesizing them. Heterotrophic microorganisms can accumulate these polymers through the catabolism of sugars or other organic molecules, de novo synthesis of fatty acids, and beta-oxidation. Conversely, autotrophic organisms can obtain the monomers necessary for PHAs from CO_2_.

### 3.1. Autotrophic Biosynthesis

Several microorganisms are capable of synthesizing PHAs using CO_2_. Through the uptake of carbon dioxide, certain bacteria, cyanobacteria, and algae can generate organic compounds via the Calvin–Benson–Bassham (CBB) cycle, with ribulose-1,5-bisphosphate carboxylase/oxygenase (RuBisCo) being the principal enzyme in this pathway [59] (Figure 2). This pathway is found in plants, algae, cyanobacteria, and some bacteria, such as *Cupriavidus necator* [59,60,61]. Although the production of PHAs by heterotrophic bacteria is 150–200 times higher than that by cyanobacteria, their low oxygen and nutrient requirements (e.g., wastewater and inorganic nitrogen and phosphorus) reduce the final cost of PHAs [62]. Both microalgae and cyanobacteria utilize sunlight as an energy source. CO_2_ uptake can occur through passive transport across the cell membrane to the chloroplast or carboxysome, where it is converted to glyceraldehyde-3-phosphate (GAP) through the CBB cycle. Alternatively, active transport of HCO_3_^−^ can occur, which involves ATP expenditure, or co-transport with anions such as Na⁺ (Figure 2A,B). In this latter case, a carbonic anhydrase (CA) is required to convert bicarbonate into carbon dioxide [63]. The GAP produced can be used for the synthesis of glucose-6-phosphate (G6P), acyl-CoA—ultimately leading to the production of free fatty acids (FFA)—or can be catabolized to pyruvate and subsequently to acetyl-CoA, which serves as a precursor for polyhydroxybutyrate (PHB) biosynthesis (Figure 2) [60,61].

Cyanobacteria such as *Synechococcus subsalsus* and *Spirulina* sp. *LEB-18* are capable of accumulating PHAs under nitrogen deficiency (Table 3). Chemical analysis of these polymers reveals that they contain chains of 14 to 18 carbon atoms, a composition previously unreported in other microorganisms [64]. In *Synechococcus*, Rueda et al. observed that the addition of acetate to the culture medium at 1.2 g/L increased PHB production up to 26.1% of cell dry weight, whereas in *Synechocystis*, acetate did not affect polymer accumulation [65]. Simonazzi et al. (2021) [62] found that *Anabaena* sp., another mixotrophic PHB producer, accumulated 40% of its dry mass as PHB under phosphorus deficiency when supplied with 5 g/L of sodium acetate. In contrast, the absence of nitrogen, with acetate maintained at the same concentration, reduced efficiency to 22.7%. In this microorganism, sulfur limitation induces oxidative stress and affects carbon allocation, leading to the production of PHB as an intracellular carbon storage [66]. In experiments with the microalga *Scenedesmus* sp., PHA production decreased from 8.61% to 2.96% when glucose (4 g/L) was added to the medium [67]. This result suggests that the presence of external carbon sources may influence the accumulation of these polymers.

Photoautotrophic organisms are highly advantageous due to the low cost of their cultivation and maintenance. Microorganisms capable of deriving energy from inorganic sources present an intriguing alternative that has been extensively studied. *Cupriavidus necator* is a facultative chemolithotroph that utilizes the CBB cycle to convert CO_2_ into GAP, an intermediate in the Entner–Doudoroff pathway that ultimately generates acetyl-CoA (Figure 2C) [94]. This bacterium possesses hydrogenases that can use molecular hydrogen as an inorganic energy source, producing ATP and reducing power in the form of NADH [59]. Additionally, this strain exhibits a heterotrophic metabolism, utilizing amino acids (e.g., alanine, valine, leucine, isoleucine, methionine, proline, phenylalanine, tryptophan, serine, threonine, glutamine, tyrosine, and histidine) as carbon sources for growth and PHA synthesis in nitrogen-free and phosphate-limited media. Leucine and isoleucine have been identified as the most efficient feedstocks, yielding close to 50% of dry cell mass, with lower yields observed when phosphate and magnesium were supplied to the medium [95].

In all cases, whether in microalgae, cyanobacteria, or chemolithotrophs, the enzyme acetyl-CoA acetyltransferase (PhaA) catalyzes the formation of acetoacetyl-CoA from two molecules of acetyl-CoA, which is then used by PhaB (an acetoacetyl-CoA reductase) to produce (R)-3-hydroxybutyryl-CoA. In the final step, poly-(3-hydroxyalkanoate) polymerase synthesizes the PHAs [59,96] (Figure 2D).

### 3.2. Heterotrophic Biosynthesis

Although some organisms are capable of utilizing inorganic carbon, numerous microorganisms with heterotrophic metabolism are capable of accumulating a diverse range of PHAs with significant chemical variability.

Methane can serve as both a carbon and energy source for methanotrophs, some of which are capable of accumulating PHAs [97]. These microorganisms employ PHAs as a survival strategy during periods when methane (CH_4_) is absent [98]. Typically, these strains are Gram-negative bacteria and are classified into two groups based on the catabolic pathways utilized for methane uptake: Type I, which uses the ribulose monophosphate pathway, and Type II, which employs the serine pathway (Figure 3) [98]. The initial step in both types of methanotrophic strains involves a methane monooxygenase (MMO) complex. Two types of MMO are recognized: a copper-containing particulate MMO (pMMO) bound to the membrane and a soluble MMO (sMMO) [99]. Under normal conditions, pMMO is the predominant enzyme, whereas sMMO is expressed in conditions of low copper availability [98]. Whereas the biochemistry, structure, and mechanism of sMMO have been extensively studied [98], the structure and activity of pMMO are more recent discoveries [100]. In both types of methanotrophs, MMO generates methanol, which is further processed through different metabolic pathways depending on the methanotroph type (Figure 3). Type I strains convert methanol into methanal (formaldehyde), which in the ribulose monophosphate (RuMP) cycle generates GAP. GAP can subsequently be converted into pyruvate and, ultimately, acetyl-CoA through glycolysis or alternative pathways. Type II strains, on the other hand, first convert methanol into formic acid, requiring additional steps to produce 5,10-methylene-tetrahydrofolate (5,10-CH_2_-THF), a key intermediate in one-carbon metabolism. Through the serine cycle, pyruvate and acetyl-CoA are then generated [101].

Some microorganisms can synthesize PHA precursors de novo from simple carbon sources such as glucose or glycerol (Table 3) [102]. For example, *Pseudomonas putida* KT2440 can convert sugars, such as glucose or fructose, and small alcohols like glycerol into acetyl-CoA. This acetyl-CoA is used in fatty acid biosynthesis to produce PHA precursors with variable carbon chain lengths and degrees of saturation, typically hydroxylated at the beta position [103]. Acetyl-CoA is also a substrate for citrate synthase, producing citrate, which subsequently enters the tricarboxylic acid cycle, ultimately forming succinyl-CoA. Through the action of SucD, 4hbD, and OrfZ, 4-hydroxybutyryl-CoA (4HB) is generated, which can accumulate as poly(4HB) with the help of PhaC [104]. Additionally, acetyl-CoA can be used to produce poly(3-hydroxybutyrate) (P3HB). In this pathway, PhaA catalyzes the condensation of two acetyl-CoA molecules to form acetoacetyl-CoA, which is then reduced by PhaB to produce 3-hydroxybutyryl-CoA. Polymerization of 3-hydroxybutyryl-CoA by poly(3-hydroxyalkanoate) polymerase results in the formation of P(3HB) (Figure 2D) [105].

A new synthetic pathway has been developed in *Escherichia coli* through the heterologous expression of genes dhaB1 (glycerol dehydratase), pduP (propionaldehyde dehydrogenase), and phaC1 (PHA synthase) from *Clostridium butyricum*, *Salmonella enterica*, and *Cupriavidus necator*, respectively. This recombinant strain is capable of accumulating poly(3-hydroxypropionate) from glycerol [106].

Typically, fatty acids are metabolized via the β-oxidation cycle, where each cycle reduces the carbon chain length by two carbons and produces one molecule of acetyl-CoA. This pathway generates a hydroxylated intermediate (3-hydroxyacyl-CoA), which can be polymerized to form PHAs with monomers having a chain length equal to or shorter than the original fatty acid [107]. In a recombinant strain of *Saccharomyces cerevisiae* carrying the PHA polymerase PhaC1 from *Pseudomonas aeruginosa*, the strain was able to accumulate PHAs with unsaturated monomers when grown in the presence of oleic acid (18:1 Δ9cis) or cis-10-heptadecenoic acid (17:1 Δ10cis). β-Oxidation of oleic acid generates monomers with a 14:1 chain length, whereas the other precursor yields monomers with H13:1 and H15:1 [108].

Heterotrophic strains such as *Pseudomonas putida* U can accumulate PHAs from various carbon sources without nutrient limitations [109]. Conversely, some PHA producers require nutrient deficiencies (e.g., nitrogen, phosphorus, magnesium, or sulfur) and an excess carbon source for optimal synthesis [110], as seen in *Pseudomonas putida* KT2440 [111]. Mixed microbial consortia are also capable of synthesizing PHAs. For instance, sludges containing microbial diversity, including *Acinetobacter*, *Flavobacterium*, *Sphingobium*, *Amaricoccus*, and *Pedobacter*, demonstrate 2.3 times higher PHA accumulation under nitrogen limitation when using volatile fatty acids from wood wastes (Table 3) [91]. Additionally, in non-pure cultures, low phosphorus concentration and nitrogen limitation reduce enzymatic activity, resulting in decreased microbial growth but increased PHA content [112].

## 4. Waste as a Resource to Produce Bioplastics

The production cost of PHB is estimated to range from USD 5380 to USD 18,300 per ton of pure product. The final cost is significantly influenced by the price of raw feedstock, prompting research into using various residues and microorganisms to enhance cost efficiency [113,114]. The agri-food industry generates a wide array of waste, which, while often of poor quality or even pollutants, can be repurposed for PHA production. Utilizing these wastes for PHA synthesis can yield high value-added products (Table 3).

The **cocoa industry** generates 4.2 tons of product annually. The pods account for 75% of the weight of the fruit, resulting in 10 tons of waste per ton of dry beans [79]. *Cupriavidus necator* has been demonstrated to accumulate PHB with a performance of 58.60 ± 4.95% (cell dry mass) when utilizing alkaline cocoa hydrolysate pod shells [79]. Fruits or vegetables with unmarketable appearances or those with short shelf lives before spoilage also provide valuable raw materials. *Cupriavidus necator* has achieved PHA production values of 79.20% (*w*/*w* on dry mass) when using such substrates (Table 3) [77]. **Cheese whey**, the principal industrial waste from the dairy industry, consists of lactose (4.5–5.0% *w*/*v*), soluble proteins (0.6–0.85% *w*/*v*), a low concentration of fat (0.36%), and minerals (0.53%), with water as the major component (over 90%) [115,116]. Complex microbial populations capable of producing biohydrogen from cheese whey, while simultaneously accumulating PHAs, have been stabilized, showing high performance (Table 3) [93]. **Wine industry** waste amounts to 5 tons per hectare annually. Valorization of this waste facilitates better resource utilization and reduced environmental impact [117]. *Cupriavidus necator*, *Halomonas halophila*, and *Halomonas organivorans* have been investigated for their abilities as PHB-producing biofactories using grape sugar extract, with performances of 70.4 ± 2.4%, 57.0 ± 1.0%, and 55.4 ± 1.4% (cell dry mass), respectively [76]. Additional studies with strains of *Bacillus*, *Tepidimonas*, *Azotobacter*, and *Pseudomonas* have demonstrated the potential of wine or grape residues as raw materials for the production of P-3HB or, in some cases, other types of PHAs (Table 3) [72,87]. **Slaughterhouse waste**, characterized by high fat content, can be utilized by certain microorganisms for PHA production. *Cupriavidus necator* and *Pseudomonas oleovorans* are capable of using these wastes to accumulate scl-PHAs [78]. Recombinant strains of *Delftia acidovorans* capable of producing PHAs from corn oil, udder, lard, and tallow have been developed, showing performances of 26.72 ± 6.66%, 26.72 ± 6.66%, 39.33 ± 1.04%, and 15.33 ± 6.11% (cell dry mass), respectively [82]. **Oils** represent another fat-rich substrate. With 97% of olive oil production located in the Mediterranean, generating 2000 tons of olive oil annually, more than 30 tons of wastewater are produced from olive mills [118]. This residue, comprising carbohydrates, lipids, and volatile fatty acids, has potential as a low-cost substrate for PHA production [69]. *Bacillus amyloliquefaciens* OM81, a strain capable of accumulating up to 30.2 ± 0.3% of its dry mass as PHAs using glucose, can also utilize olive mill wastewater, achieving performance ranging from 5.6 ± 0.1% to 11.2 ± 0.3% (cell dry mass) [69]. Additionally, waste frying oils have been employed with high performance (Table 3) [119]. **Beet molasses**, a waste from the sugar production industry, can be used without pretreatment as a substrate in microbial fermentation [80]. Processing 7 tons of sugar beets generates beet molasses (0.25–0.35 tons) as a coproduct, which can be separated into sugar, betaine, and desugared beet molasses fractions [75]. *Cupriavidus necator* has been shown to grow in beet molasses, accumulating P(3HB-co-3HV) [80]. Desugared beet molasses has been utilized as a carbon source for P(3HB) production with *Bacillus megaterium* strains, achieving production ranges of 0.55 to 0.60 g of bioplastic per gram of dry cell mass [75]. *Parapedobacter* species have also demonstrated the use of molasses as an inexpensive carbon source for PHB production (Table 3) [85]. The **wood industry** generates various wastes that, while nonedible, serve as a cheap feedstock for high-value products [86,120]. Lignocellulose, a byproduct of the agricultural and forest industries, is one of the most abundant renewable resources globally [86,92]. Although wood residues require complex processing, typically performed by microorganisms with oxidases capable of degrading lignin, microbial consortia can convert the hydrolysate into PHAs (Table 3) [86,92,120].

## 5. PHAs Extraction and Purification Process

One factor contributing to the relatively high price of PHAs compared to petroleum-derived plastics is the extraction process, which constitutes approximately 30% of production costs [121]. The extraction methods reported in the literature are classified into chemical and physical methods, which are often used individually or in combination [122].

To obtain PHAs from fermentation, several processes occur in each of the four stages described for PHA production: (1) **biomass separation**, (2) **biomass pretreatment**, (3) **extraction**, and (4) **purification** (Figure 4). Extraction typically takes place at the peak of bacterial growth to optimize polymer performance [123].

### 5.1. Biomass Separation

The initial step generally involves the separation of the biomass from the culture medium. This can be accomplished through the following: (I) Centrifugation is commonly employed in both laboratory and industrial settings, including agri-food industries, utilizing continuous centrifugation [124,125,126]. (II) Filtration processes are less common on a laboratory scale but prevalent in industrial applications. These processes are often costly due to the specific porosity required for cell size [124,125,126]. (III) Sedimentation is the most economical method but is time consuming for cell harvesting. Accumulation of cells with high moisture content leads to increased energy consumption during the subsequent cell-drying stage [124,125,126].

### 5.2. Biomass Pretreatment

Once biomass is obtained, a pretreatment stage is employed to enhance biomass permeability or achieve cell rupture. The processes include the following: (I) **Lyophilization** removes cellular water while maintaining the structure of many biomolecules. This process prevents PHAs from undergoing degradative processes and facilitates water removal for subsequent extraction steps. Enhanced freeze-drying processes with buffering agents have been observed in *Bacillus* sp. JY14. While highly effective, this method is costly to scale up to industrial levels [127]. (II) **Freezing** allows for good cell preservation due to the protective effect of PHAs during the freezing process. Freezing can be performed slowly or rapidly in liquid nitrogen. Rapid freezing further improves cell preservation by preventing the formation of ice microcrystals [128]. (III) **Cell rupture** can also occur due to increased salinity in the medium or osmotic pressure through the plasma membrane. This process is economical and facilitates the removal of PHA salts due to their hydrophobic nature [129]. (IV) **Heating** processes are widely utilized in both laboratories and industries, as low temperatures facilitate the evaporation of intracellular water. The various techniques include the following: (1) **Infrared light**: The application of infrared light promotes molecular vibration, allowing water evaporation. This technique heats the biomass surface but is limited in its penetration, requiring a large surface area for homogeneous drying [130]. (2) **Hot air**: The use of hot air currents enhances evaporation performance by improving efficiency, though it incurs higher costs. Temperatures of 45, 60, and 80 °C have been employed, with 45 °C providing similar drying results to lyophilization [131]. (3) **Pressurized hot water**: This technique disrupts cells through pressure and allows the dissolution of all polar molecules, leaving an insoluble solid with hydrophobicity where PHAs are found [132].

### 5.3. Extraction

The next step involves the extraction and recovery of the polymer, which encompasses the solubilization of either the biomass or the polymer itself. Depending on the technique and the intended application of the polymer, mechanical, enzymatic, and/or chemical methods may be employed to digest the cells, followed by the isolation of PHA through centrifugation or filtration. Solvent extraction is the most prevalent technique for recovering PHAs. Certain solvents can alter the permeability of the cell membrane and selectively dissolve the polymer. Pretreatment is occasionally performed to improve solvent accessibility to the polymer. The dissolved PHA is recovered using a precipitating agent, such as ethanol or methanol, at low temperatures. The excessive use of environmentally harmful solvents is a significant concern in the design of extraction methods. Halogenated solvents, such as chloroform, are commonly used. The Soxhlet method is employed to maximize solubility and minimize solvent use [133]. However, halogenated solvents can affect PHA morphology, are expensive, and present environmental and health risks, limiting their industrial use [134]. Alternatives to halogenated solvents include alcohols, esters, amides, and ketones [135]. Although less harmful, these alternatives still pose environmental and sustainability challenges. Dimethyl carbonate (DMC) and bio-based solvents, such as ethyl acetate, are considered more environmentally friendly [136,137]. Ionic liquids are also being investigated due to their low vapor pressure and ability to dissolve substances insoluble in water [138]. Supercritical fluids, such as supercritical carbon dioxide (sCO_2_), combine gas and liquid properties, allowing for efficient PHA extraction under mild conditions without leaving harmful residues. Although promising for pharmaceutical and biomedical products, supercritical fluids are rarely used for low-cost products due to their batch operation requirements [139].

Digestion methods for PHA recovery are the most economical processes but are time-consuming for cell harvesting. High moisture content in accumulated cells leads to increased energy consumption during the cell drying stage [124,125,126]. The digestion methods are described below.

**Chemical methods:** Sodium hypochlorite, a powerful oxidant, dissolves proteins, lipids, carbohydrates, and nucleic acids, resulting in a high-purity polymer (>95%). However, this treatment significantly reduces the molecular weight. The combination of sodium hypochlorite and chloroform minimizes this degradation, achieving effective PHA separation [140,141]. Sodium hydroxide (NaOH) and potassium hydroxide (KOH) destabilize the cell membrane, facilitating PHA extraction. Previous studies reported a purity of 88.6% and a recovery performance of 96.8% using 0.05 M NaOH at 4 °C for 1 h. Compared to sodium hypochlorite, NaOH exhibited lower efficiency in recovering PHA from mixed culture systems [142,143]. Acids, such as sulfuric acid, can break down cellular matter and release PHA. Sulfuric acid is a cost-effective and efficient process for extracting pure polymers without significant degradation, with a cost of 1.11 euros/kg and low CO_2_ emissions [144]. Surfactants, such as sodium dodecyl sulfate (SDS), increase cell membrane permeability and form micelles with phospholipids, releasing PHA granules. Although effective, surfactants are generally used in combination with other chemical or enzymatic treatments to achieve high purities. Low concentrations of SDS (0.025–0.2%) are suitable for the process, and biodegradable surfactants are considered sustainable alternatives [145].

**Enzymatic methods:** Enzymatic cocktails, including proteases, nucleases, and lysozymes, are used to lyse and digest non-PHA cellular components. Examples include the use of alcalase, SDS, and EDTA for extracting mcl-PHAs [146]. Bromelain, used at 50 °C and pH 9, yielded 88.8% purity [147]. Commercial enzymes have also been evaluated, demonstrating high recovery and purity with reduced costs.

**Biological agents:** Innovative methods for PHA recovery include the use of whole organisms that digest bacterial cellular matter, leaving PHA intact, although with long recovery times [148]. To reduce costs, bacterial and yeast strains have been engineered to secrete fermentation products accumulated in the cytoplasm. Bacteriophage lysis genes induce cell disruption at programmed times based on culture conditions, using strategies such as temperature changes and magnesium concentrations. Modified strains of *Alcanivorax borkumensis* can secrete extracellular PHA when grown on alkanes [70].

**Cell disruption methods:** Microbead milling is a common method for cell disruption and recovery of intracellular products, such as DNA, enzymes, and recombinant proteins. This process utilizes shear forces generated by glass microspheres to rupture cell walls. It has been demonstrated to be effective in completely rupturing *A. latus* cells in eight passes, regardless of biomass concentration [149]. Grinding combined with SDS treatment has achieved 100% recovery and 94% polymer purity in 2 h [150]. High-pressure homogenization (HPH) is suitable for large-scale applications in separation and purification (DSP) processing. This method applies high shear forces through a narrow space to disintegrate cells. The efficacy of HPH depends on biomass concentration, with better results obtained with medium concentrations in *Methylobacterium* cells [149]. A 98% performance and 95% purity of P(3HB) have been achieved with this technique using HPH combined with SDS [84]. This method can achieve disintegrations of over 90% even without pretreatment, which is promising for large-scale applications. However, the release of DNA can increase viscosity, which can be managed with thermal treatments, hypochlorite, or commercial nucleases. Cell disruption by ultrasound utilizes high-frequency acoustic waves converted into mechanical oscillations to break cells. This method has been used as a pretreatment in PHA recovery from various bacterial species. Ultrasound-assisted processes have been developed in combination with chemical methods to produce P(3HB) with specific properties [151]. Ultrasound and glass beads have been used as pretreatment for extraction with non-halogenated solvents [152]. Grinding with microspheres and ultrasound, adjusting pH, and using surfactants and coagulants for the recovery of PHAs is a patented process due to its high efficiency and low cost [153]. Gamma irradiation has also been explored as a method of cell disruption. Irradiation of wet biomass with doses of 5–40 kGy facilitates PHA extraction, improving polymer properties such as molecular weight and tensile strength [154].

**Cell fragility** is influenced by pretreatment methods (thermal, pH, and osmotic pressure) and the use of solvents, chemicals, and enzymes to increase cell wall fragility and release PHA granules. Factors such as the high content of PHAs in microorganisms, reaching 60–80% of dry cell matter, also contribute to their fragility, facilitating cell disruption [155]. Supplementation with fish peptone in *Azotobacter vinelandii* promotes the formation of osmotically sensitive cells, which can be lysed with simple alkaline treatment [71]. Cultivation of *B. flexus* in inorganic media reduces cell wall robustness, facilitating lysis [74]. Other methods include osmotic lysis in the presence of alkali or detergent for halophilic bacteria and the use of osmotic pressure combined with chemicals and solvents to recover P(3HB-co-3HV) from *Haloferax mediterranei* and recombinant *E. coli* cells [68,156,157].

### 5.4. Purification

After the extraction of PHA, purification follows as a crucial step. This stage is particularly critical for applications in the medical field. Contaminants vary depending on the extraction method used: lipids and dyes are coextracted with nonpolar solvents, whereas proteins are typically found following the chemical digestion of the biomass. For medical applications, PHA must have endotoxins, such as lipopolysaccharides (LPS) from Gram-negative bacteria, removed by techniques such as repeated dissolution and precipitation [158]. In *Cyanobacteria*, pigments can appear in PHB, imparting undesirable properties [159].

PHA recovery involves multiple operations that are influenced by several factors. The selection of purification methods must consider polymer quality requirements, such as molecular weight and endotoxin presence, which are particularly critical for medical applications [160,161,162]. The type of PHA (e.g., mcl-PHA or P(3HB)) and the cellular content affect the choice of extraction and purification methods [163]. Additionally, factors such as cell density, microorganism type, and cultivation conditions impact the efficiency of the process. From an environmental perspective, life cycle assessment of PHA production has demonstrated varied results, highlighting the importance of considering differences in substrates and energy systems [152,164]. Specific studies have compared purification methods, such as alkaline treatment, surfactant-hypochlorite treatment, and solvent extraction, emphasizing differences in costs and environmental impact. Alkaline treatment has been shown to provide favorable results both economically and environmentally [165].

## 6. Conclusions

The production of PHAs from agro-industrial waste and various microorganisms represents a promising and sustainable alternative to petroleum-derived plastics. Utilizing waste materials such as cocoa shells, whey, wine, wood, and beet molasses as nutrient sources for the fermentation of microorganisms capable of synthesizing PHAs offers an effective strategy to reduce costs and minimize environmental impact by reusing waste products. At the molecular level, PHA production involves complex and varied biological mechanisms. Heterotrophic microorganisms synthesize PHAs from sugars and other organic molecules through biosynthetic pathways, including the β-oxidation of fatty acids and the synthesis of acetyl-CoA. Conversely, autotrophic organisms, such as certain bacteria, cyanobacteria, and microalgae, generate the monomers required for PHA biosynthesis from CO_2_ via the CBB cycle, utilizing ribulose-1,5-bisphosphate carboxylase/oxygenase (RuBisCO). Whereas *C. necator* can utilize both inorganic and organic sources for PHA production, other organisms such as cyanobacteria (*Synechococcus subsalsus* and *Spirulina* sp.) accumulate PHAs under specific nutritional deficiency conditions. The presence of external carbon sources can negatively affect PHA accumulation in certain eukaryotic microorganisms, such as *Scenedesmus* sp., where PHA production is reduced by the addition of glucose. Conversely, other carbon sources, such as acetate, do not affect some eukaryotic microorganisms and may increase PHA levels in others. Methanotrophs, which utilize methane, have also been identified as potential PHA producers. Additionally, new biosynthetic pathways have been developed in recombinant microorganisms like *E. coli*, capable of synthesizing novel PHAs with new monomers from less common precursors.

## Figures and Tables

**Figure 1 microorganisms-12-02028-f001:**
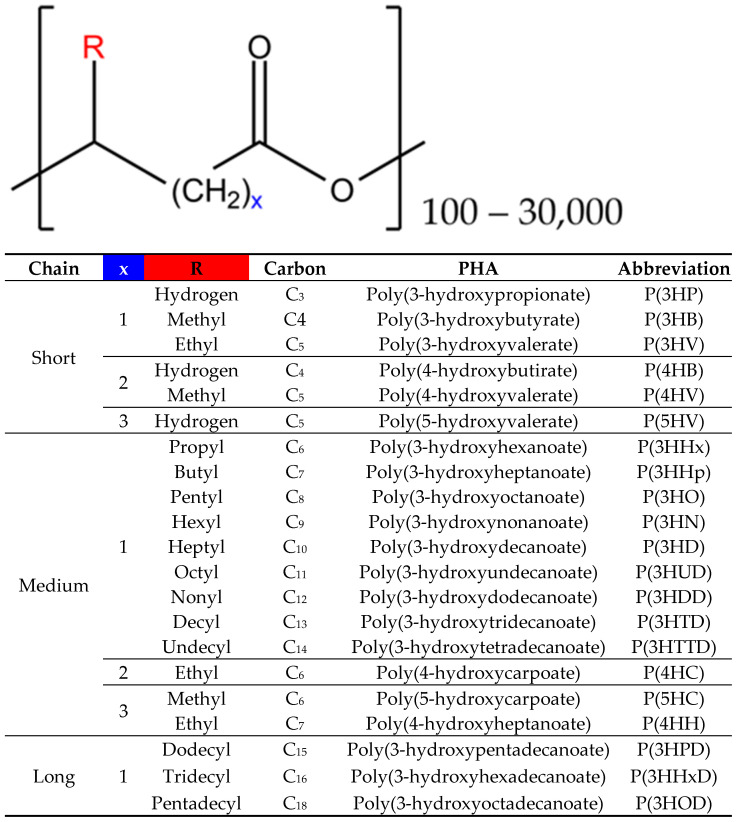
The general structure of PHAs is represented as follows: R denotes one of the radicals shown in the table, n represents the number of repeats of the monomer(s) (which can be arranged in tandem or randomly), and x signifies the number of CH_2_ groups within the main chain of each monomer. For the classification of PHAs, the total number of carbons present in the monomer is used, calculated as (R + x + 2).

**Figure 2 microorganisms-12-02028-f002:**
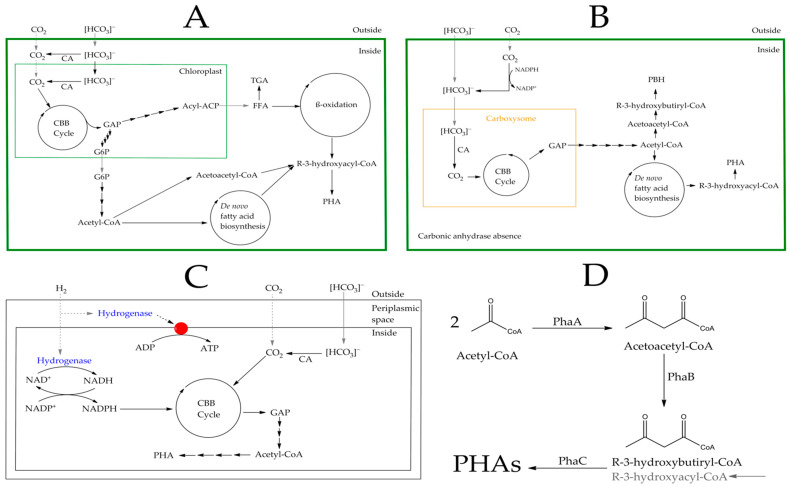
A simplified scheme of PHAs biosynthesis from CO_2_ in microalgae (**A**), cyanobacteria (**B**), and *Cupriavidus necator* as an example of a chemolithotroph (**C**) is presented. CA: carbonic anhydrase; TGA: triacylglycerides; FFA: free fatty acids; GAP: glyceraldehyde-3-phosphate; G6P: glucose-6-phosphate; PHA/PHAs: polyhydroxyalkanoates; CBB cycle: Calvin–Benson–Bassham cycle. Gray arrows indicate CO_2_ or bicarbonate capture, dashed lines represent passive diffusion, and solid arrows denote transport that may involve transporter proteins and cofactors. (**D**) Biosynthesis of polyhydroxyalkanoates from acetyl-CoA is depicted, with PhaA: acetyl-CoA acetyltransferase; PhaB: acetoacetyl-CoA reductase; PhaC: poly-(3-hydroxyalkanoate) polymerase. Gray shading illustrates the input of acyl-CoA other than *R*-3-hydroxybutyryl-CoA into the synthesis of PHAs.

**Figure 3 microorganisms-12-02028-f003:**
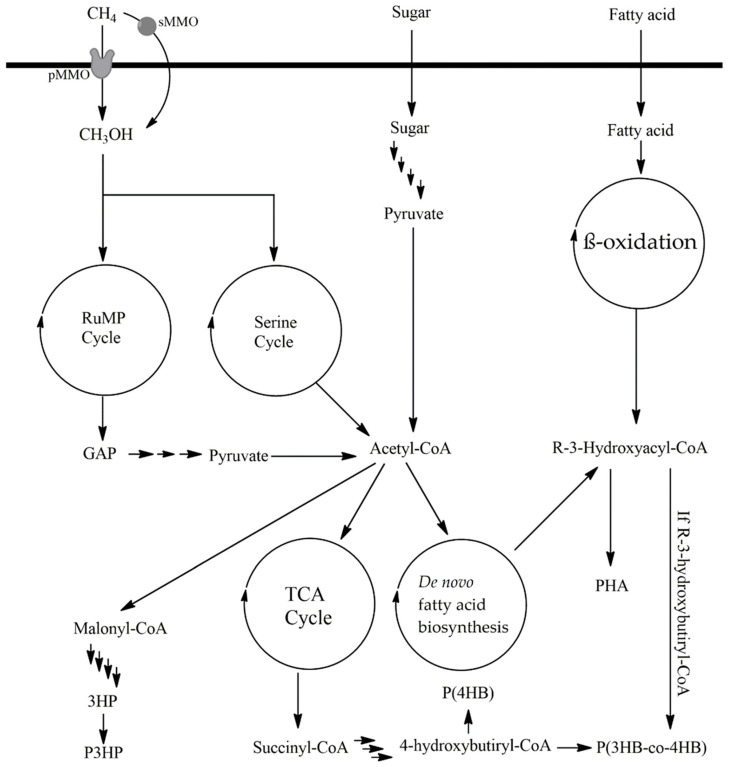
General Scheme of PHAs Synthesis in Heterotrophic Microorganisms. GAP: glyceraldehyde-3-phosphate; 3HP: 3-hydroxypropionate; 3HB: 3-hydroxybutyrate; 4HB: 4-hydroxybutyrate; P(3HP): poly(3-hydroxypropionate); P(4HB): poly(4-hydroxybutyrate); PHA: polyhydroxyalkanoate; P(3HB-co-4HB): poly(3-hydroxybutyrate-co-4-hydroxybutyrate); TCA cycle: tricarboxylic acid cycle; CH_4_: methane; CH_3_OH: methanol; pMMO: particulate methane monooxygenase; sMMO: soluble methane monooxygenase; RuMP cycle: ribulose monophosphate cycle.

**Figure 4 microorganisms-12-02028-f004:**
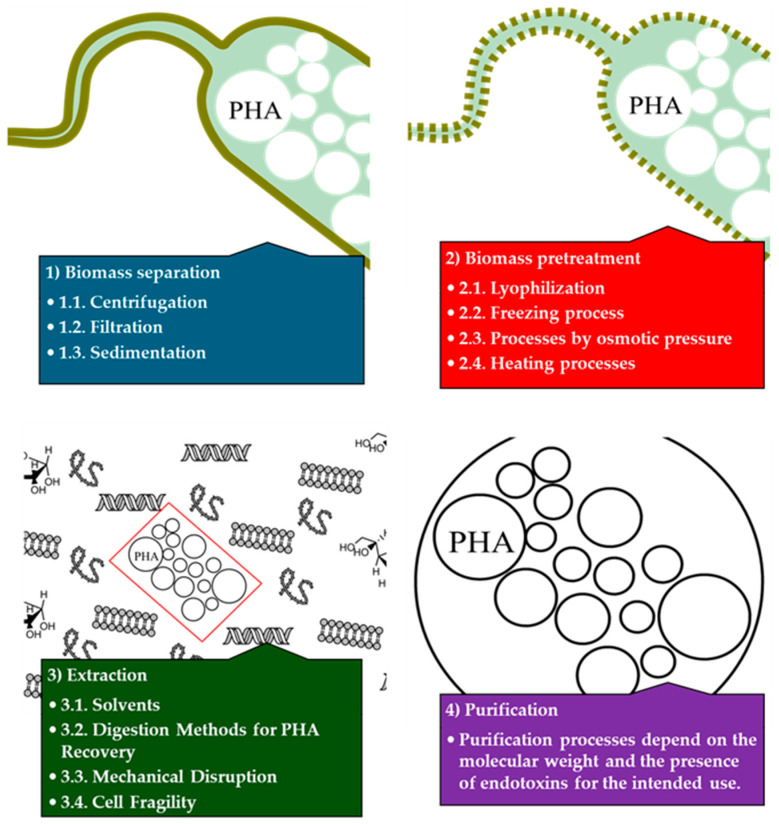
A schematic diagram illustrating the steps required to obtain PHAs generated by microorganisms is provided. All available options for each step are presented.

**Table 1 microorganisms-12-02028-t001:** Applications, useful life, and degradation time of different petrochemical plastics are compared to polyhydroxyalkanoates (PHAs). * The degradation time varies depending on the shape and thickness of the plastic object, its application, and the environment in which it is located (e.g., land, sea, temperature, etc.) [9,10,11,12,13,14].

Plastic	Applications	Usage Time	Degradation Time *
PET (Polyethylene terephthalate)	Bottles and other plastic containers	1–3 years	500–1000 years
HDPE (high-density polyethylene)	Pipelines, bottles	5–35 years	250–5000 years
LDPE (low-density polyethylene)	Plastic wrappers and bags	1–3 years	150 years
PVC (Polyvinyl chloride)	Pipelines and other uses in construction	35 years	>1000 years
PP (Polypropylene)	Textiles, packaging, automotive components	5–15 years	50–800 years
PHAs (Polyhidroxyalkanoates)	Bags, packaging, medical implants	-	<1 year

**Table 2 microorganisms-12-02028-t002:** Properties and applications of different PHAs on the size of their side chain (R).

Classification	Properties	Applications	Examples	References
Short-chain PHAs (4–5 carbons)	Fragile High melting point Greater biodegradability Biocompatibility Rigidity High crystallinity	Biofuel production Tissue engineering Drugs encapsulation	Poly(3-hydroxybutyrate) Poly(3-hydroxyvalerate)	[26,29,30,31]
Medium-chain PHAs (6–14 carbons)	More elastic Low melting point Biodegradability Biocompatibility Semi-crystalline or amorphous	Fertilizer encapsulation Adhesives Coatings Soft tissue engineering	Poly(3-hydroxyoctanoate) Poly(3-hydroxyhexanoate)	[32,33,34,35]
Long-chain PHAs (>14 carbons)	Elastic Low melting point Low glass transition temperature Low crystallinity Low tensile strength	Packaging materials	Poly(3-hydroxyhexadecanoate)	[36]

**Table 3 microorganisms-12-02028-t003:** PHA-producing microorganisms, carbon sources, pretreatments, type of polymer accumulated and performances. PHAs and PHB indicate polymers whose monomeric composition has not been identified. 3HB, 3-hydroxybutyrate; 4HB, 4-hydroxybutyrate; 3HV; 3-hydroxyvalerate; 3HH, 3-hydroxyhexanoate; 3HO, 3-hydroxyoctanoate; 3HD, 3-hydroxydecanoate; 3HDD, 3-hydroxydodecanoate. CDM: cell dry mass, Yield: g of PHA per gram of substrate, n.d.: not determined.

Phylum	Microorganism	Carbon Source	Treatment	PHA Type	PHA Concentration	% PHA (CDM)	Yield	Reference
Archaea	*Haloferax mediterranei* DSM 1411	Vinasse (25% *V*/*V*)		P(3HB-co-3HV)	19.7 g/L	70%	0.87	[68]
Bacteria	*Klebsiella oxytoca*	Glucose (20 g/L)		PHAs	0.08 ± 0.21 g/L	22.6 ± 0.1%	n.d.	[69]
	*Alcanivorax borkumensis* SK2	Pyruvate (1.5% *w*/*V*)		P(3HB)	6.5 ± 1.2 mg/L	n.d.	n.d.	[70]
		Octadecane (1.5% *w*/*V*)		P(3HH-co-3HO-co-3HD-co-3HDD)	18.0 ± 3.8 mg/L	n.d.	n.d.	
		Pyruvate (1.5% *w*/*V*)	Genetically modified strain super-producer of PHAs	P(3HB-co-3HV)	112 ± 16.8 mg/L	n.d.	n.d.	
		Octadecane (1.5% *w*/*V*)		P(3HH-co-3HO-co-3HD-co-3HDD)	2.560 ± 165.1 mg/L	n.d.	n.d.	
	*Azotobacter vinelandii*	Fish peptone (0.1% *w*/*V*)		P(3HB)	14–25 g/L	74–86%	0.29–0.65	[71]
		Sugars from grape residues (20 g/L)		P(3HB)	n.d.	37.7 ± 2.15%	0.07 ± 0.01	[72]
	*Bacillus amyloliquefaciens*	Glucose (20 g/L)		PHAs	0.14 ± 0.61 g/L	30.2 ± 0.3%	n.d.	[69]
		Olive mill wastewater (25% *V*/*V*)		PHAs	0.06 ± 0.3 g/L	5.6 ± 0.1%	n.d.	
		Olive mill wastewater (50% *V*/*V*)		PHAs	0.18 ± 0.4 g/L	5.4 ± 0.4%	n.d.	
		Olive mill wastewater (75% *V*/*V*)		PHAs	0.14 ± 0.2 g/L	11.2 ± 0.3%	n.d.	
		Olive mill wastewater (100% *V*/*V*)		PHAs	0.16 ± 0.1 g/L	7.6 ± 0.4%	n.d.	
	*Bacillus cereus*	Glucose (20 g/L)		PHAs	0.07 ± 0.08 g/L	18.9 ± 0.0%	n.d.	[69]
		Glucose (15 g/L)		PHB	1.91 ± 0.1 g/L	87.2%	n.d.	[73]
	*Bacillus flexus*	Sucrose (0.2 g/L)		P(HB-co-HV)	1.0 ± 0.42 g/L	47.67 ± 4.04%	n.d.	[74]
		Sucrose (0.2 g/L)	Yeast extract supplementation (2.5 g/L)	P(HB-co-HV)	1.6 ± 0.28 g/L	22.00 ± 15.59%	n.d.	
		Sucrose (0.2 g/L)	Beef extract supplementation (2.5 g/L)	P(HB-co-HV)	0.7 ± 0.21 g/L	42.33 ± 2.89%	n.d.	
		Sucrose (0.2 g/L)	Peptone supplementation (5 g/L)	P(HB-co-HV)	1.6 ± 0.28 g/L	30.5 ± 10.61%	n.d.	
	*Bacillus megaterium uyuni* S29	Desugarized sugar beet molasses (20% *w*/*w*)	Two different batches of beet molasses	P(3HB)	9.2 ± 0.05 g/L 10.2 ± 0.04 g/L	55 ± 0.61% 60 ± 0.08%	n.d.	[75]
	*Bacillus thioparans*	Glucose (20 g/L)		PHAs	0.14 ± 0.13 g/L	27.5 ± 0.1%	n.d.	[69]
	*Bacillus thuringiensis*	Glucose (20 g/L)		PHAs	0.03 ± 0.14 g/L	13.6 ± 0.1%	n.d.	[69]
	*Cupriavidus necator*	Grape sugar extract (20 g/L)		PHB	1.9 ± 1.5 g/L	47.2 ± 2.1%	n.d.	[76]
		Spoilage dates	Equivalent to 15.5 g/L glucose and 15.5 g/L fructose	P(3HB)	0.93 ± 0.10 g/L	73.20 ± 4.67%	0.049	[77]
		Bacon rind (3% *w*/*V*)	Finely minced	P(3HB-co-3HV)	n.d.	11 ± 2%	n.d.	[78]
		Udder (3% *w*/*V*)	Finely minced	P(3HB-co-3HV)	n.d.	55 ± 13%	n.d.	
		Tallow (3% *w*/*V*)	Finely minced	P(3HB-co-3HV)	n.d.	23 ± 2%	n.d.	
		Waste oil (3% *w*/*V*)		P(3HB-co-3HV)	n.d.	21 ± 7%	n.d.	
		Enzymatic digested waste oil (3% *w*/*V*)		P(3HB-co-3HV)	n.d.	81 ± 17%	n.d.	
		Cocoa Pod Residue	30 mL/L of alkaline pretreated cocoa pod husks	P(3HB)	n.d.	51.30 ± 2.83%	n.d.	[79]
		Fructose 20 g/L		PHB	5.5 ± 0.5 g/L	61.6 ± 1.2%	n.d.	[76]
		Grape sugar extract (20 g/L)		PHB	1.9 ± 1.5 g/L	47.2 ± 2.1%	n.d.	
		Purified grape seeds oil (20 g/L)		PHB	6.4 ± 1.9 g/L	76.8 ± 5.8%	n.d.	
		Spent coffee grounds oil (20 g/L)		PHB	6.5 ± 0.7 g/L	65.3 ± 1.0%	n.d.	
		Waste fried sunflower oil (20 g/L)		PHB	6.1 ± 0.1 g/L	70.4 ± 2.4%	n.d.	
		Hydrolysates of beet molasses	Phosphorous addition	P(3HB-co-3HV)	n.d.	80%	n.d.	[80]
		Beer brewery wastewater containing maltose		P(3HB)	n.d.	14–79%	0.01–0.33	[81]
	*Delftia acidovorans*	Myristic acid (2% *w*/*V*)		P(3HB-co-4HB-co-3HV)	n.d.	42.03 ± 9.06%	n.d.	[82]
		Oleic acid (2% *w*/*V*)		P(3HB-co-3HV)	n.d.	22.26 ± 6.85%	n.d.	
		Stearic acid (2% *w*/*V*)		PHAs	n.d.	0.26 ± 0.02%	n.d.	
		Palmitic acid (2% *w*/*V*)		P(3HB-co-4HB-co-3HV)	n.d.	32.22 ± 5.26%	n.d.	
		Corn oil (2% *w*/*V*)	Recombinant strain carring *lipH* and *lipC* from *Pseudomonas stutzeri*	P(3HB-co-4HB-co-3HV)	n.d.	26.72 ± 6.66%	n.d.	
		Udder (2% *w*/*V*)	Recombinant strain carring *lipH* and *lipC* from *Pseudomonas stutzeri*	P(3HB-co-4HB-co-3HV)	n.d.	26.72 ± 6.66%	n.d.	
		Lard (2% *w*/*V*)	Recombinant strain carring *lipH* and *lipC* from *Pseudomonas stutzeri*	P(3HB-co-4HB-co-3HV)	n.d.	39.33 ± 1.04%	n.d.	
		Tallow (2% *w*/*V*)	Recombinant strain carring *lipH* and *lipC* from *Pseudomonas stutzeri*	P(3HB-co-4HB-co-3HV)	n.d.	15.33 ± 6.11%	n.d.	
	*Halomonas alkaliantarctica*	Cheese whey mother liquor (110 g/L)		P(3HB-co-3HV)	0.42 g/L	20.1%	n.d.	[83]
	*Halomonas halophila*	Glucose (20 g/L)		P(3HB)	3.7 ± 0.6 g/L	72.5 ± 0.9%	n.d.	[76]
		Grape sugar extract (20 g/L)		P(3HB)	1.8 ± 0.6 g/L	57.0 ± 1.0%	n.d.	
	*Halomonas organivorans*	glucose (20 g/L)		P(3HB)	3.9 ± 8.2 g/L	66.0 ± 9.2%	n.d.	[76]
		Grape sugar extract (20 g/L)		P(3HB)	2.1 ± 0.4 g/L	55.4 ± 1.4%	n.d.	
	*Methylobacterium* sp V49	Methanol (0.5%)		P(3HB)	0.08 g/L	11%	n.d.	[84]
		Glucose (15 g/L)		P(3HB)	0.80 g/L	53%	n.d.	
		Fructose (15 g/L)		P(3HB)	0.3 g/L	25%	n.d.	
		Sucrose (15 g/L)		P(3HB)	0.5 g/L	28%	n.d.	
		Lactose (15 g/L)		P(3HB)	0.11 g/L	15%	n.d.	
	*Parapedobacter* sp. ISTM3	Molasses (5% *V*/*V*)	Nitrogen-limiting medium	PHB	0.47 ± 0.02 g/L	55.62 ± 0.44%	n.d.	[85]
	*Pseudomonas oleovorans*	Bacon rind (3% *w*/*V*)		co-polymers containing 3HV	n.d.	8 ± 2%	n.d.	[78]
		Udder (3% *w*/*V*)		co-polymers containing 3HV	n.d.	15 ± 3%	n.d.	
		Tallow (3% *w*/*V*)		co-polymers containing 3HV	n.d.	13 ± 2%	n.d.	
		Waste oil (3% *w*/*V*)		co-polymers containing 3HV	n.d.	21 ± 6%	n.d.	
		Enzymatic digested waste oil (3% *w*/*V*)		co-polymers containing 3HV	n.d.	76 ± 14%	n.d.	
	*Pseudomonas putida* NX-1	Kraft lignin (10 g/L)		mcl-PHAs	114.16 mg/L	37.64%	n.d.	[86]
	*Pseudomonas* sp. PhDV1	Grape pomace (1% *V*/*V*)	Δ*phaZ* strain	P(3HB)	n.d.	16%	n.d.	[87]
Cyanobacteria	*Anabaena* sp.	CO_2_ and acetate (1 g/L)	Mixotrophic growth phosphorous limitation	P(3HB)	0.07 g/L	37.4 ± 3.0%	n.d.	[62]
	*Aulosira fertilissima*	Acetate + Citrate + CO_2_		P(3HB)	31.7 ± 0.23 mg/L	6.4 ± 0.05%	n.d.	[88]
		Citrate (0.5% *w*/*V*)	Dark conditions and phosphorous deficiency	P(3HB)	73.1 ± 0.82 mg/L	44.6 ± 0.19%	n.d.	
		Acetate (0.5 *w*/*V*)	Phosphorous deficiency	P(3HB)	160.1 ± 1.02 mg/L	77.2 ± 1.89%	n.d.	
		Citrate (0.26% *w*/*V*) + acetate (0.28% *w*/*V*)	5.58 mg/L K_2_HPO_4_	P(3HB)	n.d.	85.1 ± 0.94%	n.d.	
	*Nostoc muscorum*	CO_2_ + citrate (0.012 mM)		PHB	n.d.	26.37%	n.d.	[89]
	*Synechocystis* sp.	NaHCO_3_ (5 mM)		PHAs	n.d.	n.d.	n.d.	[90]
Microalgae	*Scenedesmus* sp.	Glucose (1–4 g/L)		PHAs	0.007–0.239 g/L	0.831–29.92%		[67]
Mixed microbial consortia	Mixed microbial consortia	Wood hydrolysis-enzymatic hydrolysate	Anaerobiosis; fatty acid proportion even/odd = 88:12)	PHAs	n.d.	50.3 ± 0.7%	n.d.	[91]
			Anaerobiosis; fatty acid proportion (even/odd = 63:37)	PHAs	n.d.	44.9 ± 0.5%	n.d.	
			Anaerobiosis; fatty acid proportion (even/odd = 54:46)	PHAs	n.d.	44.5 ± 0.6%	n.d.	
			Anaerobiosis; fatty acid proportion (even/odd = 48:52)	PHAs	n.d.	44.7 ± 0.8%	n.d.	
		Rubber wood hydrolysate (1 g/L)	Xylose supplemented up to 1:1 ratio with glucose from hydrolysate	P(3HB-co-3HV)	n.d.	16.9–43.6%	n.d.	[92]
		Cheese whey		P(3HB)	n.d.	55.1–62%	n.d.	[93]

## Data Availability

All relevant data are included in the manuscript.

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
