# Peer review of "A Review of Polyhydroxyalkanoates: Characterization, Production, and Application from Waste"

_microorganisms, 2024, doi:10.3390/microorganisms12102028_

Round 1

Reviewer 1 Report

Comments and Suggestions for Authors

The report is attached.

Author Response

Comments 1: The manuscript offers a comprehensive review of the developments in the characterisation, production and application of polyhydroxyalkanoates. The manuscript is well written and organised; however, the novelty and contribution of the work to the field are not immediately apparent. A substantial number of review manuscripts on polyhydroxyalkanoates already exist, including: https://www.sciencedirect.com/science/article/abs/pii/S0141813021015804, https://www.mdpi.com/2073-4360/13/10/1544, https://www.sciencedirect.com/science/article/abs/pii/S0960852402002122, https://www.sciencedirect.com/science/article/abs/pii/S0921344923003014, https://www.sciencedirect.com/science/article/abs/pii/S096085242101350X, https://www.frontiersin.org/journals/bioengineering-andbiotechnology/articles/10.3389/fbioe.2021.624885/full, and others.

Response 1: Thank you very much for your feedback. We would like to mention that this manuscript is an update to the works you referred to, and it also introduces a wide variety of microorganisms, including photosynthetic ones. A key novelty of this review is the comprehensive coverage of different microorganisms, alongside the use of waste materials and extraction methods.

Comments 2: Point 1: It is recommended that, in the Introduction, the authors explicitly delineate the novel aspects of their work.

Response 2: The importance and contribution of this work has been added as indicated by the reviewer.

Comments 3: It is recommended that Section 4 (Waste as a Resource to Produce Bioplastics) be reorganised into a single section, without subdivision into subsections.

Response 3:  We appreciate the reviewer's suggestion. Section 4 is modified by eliminating the headings and leaving only one section. Bold type is maintained when mentioning each of the residues. We believe that this allows a better search of the data for the readers.

Comments 4: It is recommended that Section 5 (PHAs extraction and purification process) be reorganised into 4 subsections (1. Biomass separation, 2. Biomass pretreatment, 3. Extraction and 4. Purification, without subdivision into labelled paragraphs.

Response 4: We have reorganized the extraction section as per the reviewer’s suggestions.

Comments 5: It would be beneficial for the authors to consider including a general section that covers the strategies that researchers have adopted in terms of genetic tools and adaptive laboratory evolution for polyhydroxyalkanoates by yeast.

Response 5: The review focuses on the production of PHAs in general terms. Some mention is made of genetically modified organisms (lines 309-313; 360-363; Table 3: Delftia acidovorans), one of which refers to the yeast Sacharomyces cerevisiae. The previous organization of the manuscript did not consider going deeper into gene editing in microorganisms (prokaryotes or eukaryotes).

Reviewer 2 Report

Comments and Suggestions for Authors

The authors have not met the necessary requirements for a review paper and have not thoroughly covered the literature. To address this, I have compiled a list of additional, more recent papers that are directly related to the field the authors are researching. These references should help provide a more comprehensive understanding of the current state of research in this area.

1.    Li, Z., Yang, J. & Loh, X. Polyhydroxyalkanoates: opening doors for a sustainable future. NPG Asia Mater 8, e265 (2016). https://doi.org/10.1038/am.2016.48
2.    Hye Min Song, Jeong Chan Joo, Seo Hyun Lim, Hye Jin Lim, Siseon Lee, Si Jae Park, Production of polyhydroxyalkanoates containing monomers conferring amorphous and elastomeric properties from renewable resources: Current status and future perspectives, Bioresource Technology, 366 (2022) 128114
3.    Malte Winnacker, Polyhydroxyalkanoates: Recent Advances in Their Synthesis and Applications, European Journal of Lipid Science and Technology, 121, Issue 11 1900101
4.    Aishwarya Pandey, Ndao Adama, Kokou Adjallé, Jean-François Blais, Sustainable applications of polyhydroxyalkanoates in various fields: A critical review, International Journal of Biological Macromolecules, 221, (2022) 1184-1201.

Page 1; Line 36: What is the source of the sentence? Only 9,6% of them have a non-petrochemical origin (bio-based, recycled or carbon capture).

Page 1; Line 39: Unnecessary dot in the middle of the sentence (These regions are also the main biobased plastic producers with 33%, 13% and 27%, of global. production, respectively)

Page 1; Line 42: There are excess letters in sentence Plastics are synthetic polymeric molecules whose characteristics make them suitable for diverse l uses

In addition to the chapters mentioned in the papaer, it would be beneficial to describe the potential applications of polyhydroxyalkanoates (PHAs) in various fields. This would provide a more complete picture of their versatility and potential impact in industries such as medicine, agriculture, and packaging, thereby highlighting the practical significance of the research.

Due to the aforementioned reasons, I believe this paper needs to be thoroughly revised and enriched with the extensive literature that exists in the scientific community in this field. After these revisions, the paper can be reconsidered for publication.

Comments on the Quality of English Language

The English language needs to be significantly improved, the sentences are long and unclear. Punctuation marks needs to be upgraded

Author Response

Comments 1:The authors have not met the necessary requirements for a review paper and have not thoroughly covered the literature. To address this, I have compiled a list of additional, more recent papers that are directly related to the field the authors are researching. These references should help provide a more comprehensive understanding of the current state of research in this area.

  1. Li, Z., Yang, J. & Loh, X. Polyhydroxyalkanoates: opening doors for a sustainable future. NPG Asia Mater 8, e265 (2016). https://doi.org/10.1038/am.2016.48
    2.    Hye Min Song, Jeong Chan Joo, Seo Hyun Lim, Hye Jin Lim, Siseon Lee, Si Jae Park, Production of polyhydroxyalkanoates containing monomers conferring amorphous and elastomeric properties from renewable resources: Current status and future perspectives, Bioresource Technology, 366 (2022) 128114
    3.    Malte Winnacker, Polyhydroxyalkanoates: Recent Advances in Their Synthesis and Applications, European Journal of Lipid Science and Technology, 121, Issue 11 1900101
    4.    Aishwarya Pandey, Ndao Adama, Kokou Adjallé, Jean-François Blais, Sustainable applications of polyhydroxyalkanoates in various fields: A critical review, International Journal of Biological Macromolecules, 221, (2022) 1184-1201.

Response 1: We thank the reviewer for the references suggested to improve the focus of the article. They have been reviewed and used to improve the manuscript. Additionally, they have served to include the applications that these polymers have as indicated below.

Comments 2: Page 1; Line 36: What is the source of the sentence? Only 9,6% of them have a non-petrochemical origin (bio-based, recycled or carbon capture).

Response 2: The subject of the sentence is plastics. The sentence is edited for better understanding. “Only 9,6% of these petrochemical polymer have a bio-based, recycled or carbon capture origin”

Comments 3: Page 1; Line 39: Unnecessary dot in the middle of the sentence (These regions are also the main biobased plastic producers with 33%, 13% and 27%, of global. production, respectively)

Response 3: Dot is deleted.

Comments 4: Page 1; Line 42: There are excess letters in sentence Plastics are synthetic polymeric molecules whose characteristics make them suitable for diverse l uses

Response 4: The remaining letter l is removed

Comments 5: In addition to the chapters mentioned in the paper, it would be beneficial to describe the potential applications of polyhydroxyalkanoates (PHAs) in various fields. This would provide a more complete picture of their versatility and potential impact in industries such as medicine, agriculture, and packaging, thereby highlighting the practical significance of the research.

Response 5: In the final part of section 2, the applicability of these PHAs in various fields has been highlighted in yellow, as suggested by the reviewer. Additionally, references to the works indicated by the reviewer have been included.

Reviewer 3 Report

Comments and Suggestions for Authors

The manuscript is an interesting review of PHA production from organic waste. The manuscript contains valuable information, however major revision is needed in my opinion:

- Table 3 has a column labelled "yield" which is very confusing and misleading. The values reported under yield belong in reality to different categories: yield, concentration and content in the microorganisms. The yield should represent the fraction of the organic matter converted into PHAs (this should be in % on a C basis or on an organic matter basis and this should be specified); the concentration represents the concentration of PHAs referred to the liquid medium (units mg or g/L); content in the microorganisms should represent the % of the total microbial mass made by PHAs. In the manuscript, these three entirely different concepts are confused and mixed together under the "yield" column, which is incorrect. I suggest using three different columns for the three concepts and giving the values for each of them, if reported in the papers;

- Similarly, in the text the concepts of yield, concentration and content in microorganisms should be distinguished and named appropriately. 

- Section 4 on wastes is interesting but should be developed further. I suggest using a table to summarise the main characteristics of each type of waste, e.g. generation rate, main components (water, proteins, lipids, etc);

- Section 5. on extraction/purification should also include more information. For example, it's important to add info on the operating conditions (mainly T and P) of the various processes, for example, which temperatures/pressures are used typically in solvent extraction?

Comments on the Quality of English Language

English language is at a satisfactory standard.

Author Response

Comments 1: Table 3 has a column labeled "yield" which is very confusing and misleading. The values reported under yield belong in reality to different categories: yield, concentration and content in the microorganisms. The yield should represent the fraction of the organic matter converted into PHAs (this should be in % on a C basis or on an organic matter basis and this should be specified); the concentration represents the concentration of PHAs referred to the liquid medium (units mg or g/L); content in the microorganisms should represent the % of the total microbial mass made by PHAs. In the manuscript, these three entirely different concepts are confused and mixed together under the "yield" column, which is incorrect. I suggest using three different columns for the three concepts and giving the values for each of them, if reported in the papers;

Response 1:  The concept of performance presented in Table 3 has been clarified, and the modifications suggested by the reviewer have been implemented. We appreciate the comment, as it has helped improve the clarity of the text. Additionally, all the data in the table has been revised, and some minor errors (modified and marked) have been corrected. We apologize for them

Comments 2:Similarly, in the text the concepts of yield, concentration and content in microorganisms should be distinguished and named appropriately. 

Response 2:  In the text, the performance metrics have been detailed according to the reviewer’s criteria, specifying whether they refer to concentration, yield, or the mass of PHAs relative to dry cell mass.

Comments 3: Section 4 on wastes is interesting but should be developed further. I suggest using a table to summarise the main characteristics of each type of waste, e.g. generation rate, main components (water, proteins, lipids, etc);

Response 3:  The chemical composition of the waste is very relevant. The main problem in this point is that in many of the articles used, the chemical characterization of the waste used is not carried out. One could resort to literature to know the composition of the different raw materials, however, the complex composition of these wastes would give a different composition to the one that has actually been used.

Comments 4: Section 5. on extraction/purification should also include more information. For example, it's important to add info on the operating conditions (mainly T and P) of the various processes, for example, which temperatures/pressures are used typically in solvent extraction?

Response 4:  Thank you for your feedback. However, we believe that it may not be necessary to include this information, as in many extraction methods, these parameters depend on the microorganism involved, which can either facilitate or hinder the extraction of PHAs. The main goal of this review is to describe all the mechanisms reported, but we cannot generalize the entire process independently of the microorganism. Therefore, we believe it is better not to include these parameters.

Round 2

Reviewer 1 Report

Comments and Suggestions for Authors

I recommend the corrected version form publication.

Author Response

Thank you very much

Reviewer 2 Report

Comments and Suggestions for Authors

The authors have responded to the reviewers' comments and have made the necessary revisions. We believe the manuscript now addresses all concerns raised and meets the required standards. Therefore, we kindly recommend to accept the Manuscript after minor editing of English language.

Comments on the Quality of English Language

Minor Editing

Author Response

We would like to inform you that the English has been thoroughly reviewed by a U.S.-based academy specialized in editing scientific articles in American English. We believe that the manuscript has improved and is now suitable for acceptance for publication.
Thank you for everything, and we look forward to your response.